# Oncogenic STAT Transcription Factors as Targets for Cancer Therapy: Innovative Strategies and Clinical Translation

**DOI:** 10.3390/cancers16071387

**Published:** 2024-03-31

**Authors:** Weiyuan Wang, Melanie Cristina Lopez McDonald, Rajashree Hariprasad, Tiara Hamilton, David A. Frank

**Affiliations:** 1Department of Hematology and Medical Oncology, Winship Cancer Institute, School of Medicine, Emory University, Atlanta, GA 30322, USA; weiyuan.wang@emory.edu (W.W.); melanie.cristina.lopez.mcdonald@emory.edu (M.C.L.M.); tiara.hamilton@emory.edu (T.H.); 2Alabama College of Osteopathic Medicine, Dothan, AL 36303, USA; hariprasadr@acom.edu

**Keywords:** STAT3, STAT5, cancer, immunity, cancer therapy

## Abstract

**Simple Summary:**

Cancer is the second leading cause of death worldwide, accounting for nearly one in six deaths. One of the key underlying factors distinguishing a cancer cell from a normal cell is the pattern of expression of genes. Proteins that regulate gene expression, called transcription factors, are abnormally regulated in cancer cells, particularly a group of proteins called STATs. This causes cancer cells to survive, proliferate excessively, and escape killing by the immune system. Since normal cells can survive adequately without full STAT function, targeting these proteins is an attractive approach for a new generation of more effective and less toxic cancer therapies. In this review, we summarize the current knowledge of STAT function in cancer and the advances and challenges in developing drugs to target them.

**Abstract:**

Despite advances in our understanding of molecular aspects of oncogenesis, cancer remains a leading cause of death. The malignant behavior of a cancer cell is driven by the inappropriate activation of transcription factors. In particular, signal transducers and activators of transcription (STATs), which regulate many critical cellular processes such as proliferation, apoptosis, and differentiation, are frequently activated inappropriately in a wide spectrum of human cancers. Multiple signaling pathways converge on the STATs, highlighting their importance in the development and progression of oncogenic diseases. STAT3 and STAT5 are two members of the STAT protein family that are the most frequently activated in cancers and can drive cancer pathogenesis directly. The development of inhibitors targeting STAT3 and STAT5 has been the subject of intense investigations in the last decade, although effective treatment options remain limited. In this review, we investigate the specific roles of STAT3 and STAT5 in normal physiology and cancer biology, discuss the opportunities and challenges in pharmacologically targeting STAT proteins and their upstream activators, and offer insights into novel therapeutic strategies to identify STAT inhibitors as cancer therapeutics.

## 1. Introduction: STAT Transcription Factors

Cancer is the second leading cause of death after cardiovascular diseases worldwide, accounting for nearly 10 million deaths in 2020, or nearly 1 in 6 deaths [1]. Despite advances in our understanding of mechanisms of cancer pathogenesis and the development of novel modes of therapy, most advanced cancers remain incurable. To develop novel therapies that have greater efficacy and less toxicity, there is an interest in identifying cellular pathways on which malignant cells, but not normal cells, are dependent. Since the phenotype of a cancer cell, including properties such as invasion and metastasis, are driven by the pattern of gene expression, there is a particular interest in identifying transcription factors that control critical cellular processes and that become activated inappropriately in cancer. One such family of transcription factors is the STAT family.

The STAT family was discovered as key mediators of cytokine signaling and interferon (IFN)-related anti-viral activity in the late 1980s and early 1990s [2,3,4]. STATs, an acronym for signal transducers and activators of transcription, comprise seven members: STAT1, STAT2, STAT3, STAT4, STAT5A, STAT5B, and STAT6. These proteins share a conserved structure and a common mechanism of action but have different functions in normal cells and tumor biology [5]. Their structure is characterized in sequence by an N-terminal domain, a coiled-coil domain, a DNA-binding domain, a linker region, a phosphotyrosine-binding Src homology 2 (SH2) domain, and a C-terminal transactivation domain. Under basal conditions, STATs are found in the cytoplasm of cells as inactive dimers. When they become phosphorylated on a single tyrosine residue towards the carboxy terminus, they undergo a conformational change, leading to reciprocal phosphotyrosine-SH2 interactions. This reveals a nuclear localization signal that allows the STAT dimers to translocate from the cytoplasm to the nucleus and to bind specific 9 or 10 base pair motifs in the regulatory regions of target genes. Although their name indicates that they activate the transcription of target genes, like most transcription factors, STATs can also repress the expression of genes in a context-dependent fashion.

STATs are commonly activated in response to cytokines, many of whose receptors are associated with one or more of the four JAK family kinases (JAK1, JAK2, JAK3, and TYK2). Hence, signaling through these transcription factors is often categorized by the shorthand “JAK-STAT” signaling. However, given the broad range of genes regulated by these seven STAT family members in different cell types, this designation over-simplifies the diversity and complexity of the role of these proteins. The designation “JAK-STAT” also does not take into consideration the fact that many tyrosine kinases other than JAK family members can phosphorylate and activate STAT proteins. This includes receptor tyrosine kinases (including the receptors for epidermal growth factor (EGF), fibroblast growth factor (FGF), hepatocyte growth factor (HGF), and others) as well as non-receptor tyrosine kinases of the SRC family and others. Not only can STATs be activated in response to cytokines and growth factors, but STATs can also be activated downstream of cell–cell and cell–matrix interactions.

STAT tyrosine phosphorylation and nuclear translocation occur within seconds of cytokine stimulation. Reflecting the critical functions of target genes regulated by STATs, they are also inactivated very rapidly by a number of mechanisms, including protein tyrosine phosphatases (PTPs), protein inhibitors of activated STAT (PIAS), and suppressors of cytokine signaling proteins (SOCS) [6]. Nuclear PTPs can dephosphorylate STAT proteins, which leads to their inactivation and subsequent transport out of the nucleus [7]. SOCS proteins function in the cytoplasm and can bind various phosphotyrosines on intracellular receptors, blocking STATs from their native docking sites [8]. PIAS proteins can bind to phosphorylated STAT dimers, thus preventing DNA recognition and STAT-mediated signaling cascades [9,10].

## 2. Inappropriate Activation of STATs in Cancer

As noted, the seven STAT family members are activated in a wide array of cell types and regulate the expression of many different genes. However, reflecting their role in directing key biological programs in response to cytokines and other extracellular cues, the target genes of STATs have certain commonalities [11]. Many STAT target genes regulate core cellular processes such as survival, proliferation, self-renewal, and differentiation. Functionally, STAT activation underlies processes such as cell motility, invasion, angiogenesis, and immune function and recognition [12,13]. Notably, STATs also regulate the expression of other transcription factors (including themselves). Thus, STAT activation leads to the initiation of multiple transcriptional programs in a coordinated way and amplifies its own response. Consequently, the tight regulation of STAT function is important for physiologic homeostasis. On the other hand, inappropriate activation of STATs, either through increased activity of upstream kinases, decreased expression or function of negative regulators, or both, can lead to significant perturbations in cellular function [12]. In fact, shortly after the discovery of these proteins, it was observed that inappropriate or constitutive phosphorylation and activation of these proteins occurred commonly in human cancer and model systems of malignancy, which can occur through various mechanisms [14,15] (Figure 1). In a variety of well-defined preclinical models that mimic various features of human cancer (summarized in Table 1), it has been found that inappropriate activation of STAT transcription factors can directly drive malignancy [16,17,18]. As described below, the inappropriate function of essentially each STAT family member, particularly STAT3 and STAT5, has been associated with human cancer.

### 2.1. STAT3

STAT3 was originally described as an “acute phase response factor (APRF)” in that it mediates the effects of interleukin-6 (IL-6) and related cytokines in response to tissue injury, inflammation, and infection. Its target genes regulate cell proliferation, differentiation, apoptosis, angiogenesis, inflammation, and immune responses [19,20] (Table 2). Constitutive or inappropriate activation of STAT3 has been reported in many types of cancers [21,22,23]. Generally, this is due to increased phosphorylation or decreased inactivation of STAT3. For example, heat shock protein 110 can directly bind to STAT3 and facilitate its phosphorylation, contributing to tumor growth in colon cancer patient samples [24]. The activated Notch1 receptor increases the level of phosphorylated STAT3, promoting cancer progression in gastric cancer [25]. Silencing of *SOCS3* causes decreased inactivation of STAT3, which also contributes to its constitutive activation and, thus, the progression of cancers such as hepatocellular carcinoma and cholangiocarcinoma in both cell and animal models [26,27,28]. Persistent STAT3 activation plays a central role in tumorigenesis [29,30]. It can promote the transcription of target genes such as BCL-6, MCL1, and MYC to promote proliferation and survival [31]. It can also activate genes such as HGF, VEGF, and HIF-1α while decreasing the expression of target genes such as IL-12 and p53 to promote angiogenesis [32]. In addition, STAT3 is widely involved in metastasis by increasing the expression of genes such as MMP2/9, Twist, and Vimentin [33].

In some forms of cancer, such as large granular lymphocytic T cell leukemia, mutations occurring in STAT3 itself lead to increased magnitude or duration of STAT3 phosphorylation [5,34,35,36]. Large granular lymphocyte (LGL) leukemia is an indolent lymphoproliferative disorder of mature T and natural killer (NK) cell neoplasms, as described in the 2016 World Health Organization classification [37,38]. There are two types of LGL leukemia: the T cell (T-LGL) and the natural killer cell (NK-LGL). Somatic gain-of-function STAT3 mutations are demonstrated in 28–75% of T-LGL leukemia and 30–48% of NK-LGL leukemia [39]. Most STAT3 mutations are detected in the SH2 domain, which drives the dimerization and activation of the STAT protein [40]. The amino acid changes result in a more hydrophobic protein surface and are associated with the phosphorylation of STAT3 and its localization in the nucleus [41]. Mutations outside the SH2 domain are rare but have been found in the DNA-binding and coiled-coil domains [42]. It has been suggested that STAT3 mutations may not be the initial trigger of the leukemic process in LGL leukemia [43]. This is supported by pre-clinical evidence that expression of a STAT3 mutant alone is not sufficient to induce LGLL in animal models and that inhibition of STAT3 restores apoptosis of LGL cells regardless of the STAT3 mutation status [39,44]. This is also supported by clinical evidence that STAT3-unmutated patients show hyperactive STAT3, but STAT3-mutated cases may remain in subclinical states for a very long time, sometimes indefinitely [43,45]. Rather, STAT3 mutations have been found to cause a higher level of transcription of survival components, thus conferring a competitive growth advantage on clonal accumulation and autoimmunity [45,46,47].

Aberrant STAT3 activation can promote oncogenesis by its cell-autonomous effects in a cancer cell [19]. Other non-malignant cells in the surrounding area can also have increased STAT3 activation since STAT3 can become activated in the tumor microenvironment in part due to the presence of cytokines that activate this protein, like IL-6 [48]. This can be of particular significance in immune cells, in which enhanced STAT3 activation can lead to decreased antigen presentation and immune effector function [49]. In innate immunity, STAT3 regulates critical steps during emergency granulopoiesis to help contain infection, restrains neutrophil production to limit inflammatory responses, and suppresses the maturation and activation of dendritic cells to induce immunosuppressive effects [50,51,52,53]. In adaptive immunity, STAT3 positively regulates an early step in B-cell development and promotes the differentiation and maturation of plasma cells [54,55]. It can also promote the proliferation and diversity of CD4+ T cells and generate stable, long-lived CD8+ memory T cells [56,57]. However, in aggregate, STAT3 activation in immune cells helps create an immunosuppressive microenvironment that is permissive to the maintenance and spread of cancer cells [58].

**Table 2 cancers-16-01387-t002:** Common direct STAT3 target genes.

Gene	Function	Status	Cell Source	References
*AKT1*	Proliferation	Upregulated	Various human cancer cells	[59]
*BATF*	Differentiation	Upregulated	Human Th17 cells	[60]
*Bcl-xL*	Anti-apoptosis	Upregulated	Human U266 cells	[14]
*BCL6*	Proliferation	Upregulated	Human Th17 cells	[60]
*MYC*	Proliferation	Upregulated	Murine Ba/F3 cells	[61]
*CCND1*	Proliferation	Upregulated	Human gastric cancer cells	[62]
*CDKN2C*	Cell cycle inhibition	Downregulated	Human Th17 cells	[60]
*CREM*	Spermatogenesis	Downregulated	Human Th17 cells	[60]
*CXCL10*	Angiogenesis, Immune escape	Downregulated	Human CD8+ T cells	[63]
*FOSL2*	Differentiation	Upregulated	Human Th17 cells	[60]
*IKZF2*	Lymphocyte development	Downregulated	Human Th17 cells	[60]
*IL6*	Immune escape	Upregulated	Murine melanoma cells	[64]
*IL10*	Immune escape	Upregulated	Human colon Carcinoma	[65]
*MMP2*	Immune escape	Upregulated	Murine melanoma cells	[66]
*MMP9*	Immune escape	Upregulated	Murine fibroblasts	[67]
*CCL5*	Immune escape	Downregulated	Murine melanoma cells	[64]
*RBPJ*	Differentiation	Upregulated	Human Th17 cells	[60]
*SMAD7*	Differentiation	Downregulated	Human Th17 cells	[60]
*STAT1*	Differentiation	Downregulated	Human Th17 cells	[60]
*STAT2*	Antiviral activity	Downregulated	Human Th17 cells	[60]
*STAT3*	Differentiation	Upregulated	Human Th17 cells	[60]
*TWIST*	Immune escape	Upregulated	Human breast carcinomas	[68]
*VEGF*	Angiogenesis, immune escape	Upregulated	Murine fibroblasts	[67]
*VIM*	Immune escape	Upregulated	Monkey kidney cells	[69]

### 2.2. STAT5

STAT5 refers to two highly related proteins, STAT5A and STAT5B, that are encoded by adjacent highly homologous genes, which likely arose through gene duplication. STAT5 transduces signals from a number of cytokines that regulate hematopoiesis at the level of hematopoietic stem cells, hematopoietic progenitor cells, and mature cell populations [70] (Table 3). STAT5 is required for cellular “fitness” in hematopoietic stem and progenitor cells, with STAT5 deficiency resulting in greatly impaired long-term multilineage repopulation capacity [71,72,73]. In natural killer cells, STAT5 helps mediate cell development, maturation, and homeostasis [74]. In adaptive immune cells, STAT5 plays a critical role in differentiation and development [75,76] and promotes the differentiation of B cells [77]. It also promotes cell proliferation and active cellular uptake of carnitine in CD4+ T cells and enhances BCL2 expression and cell survival of CD8+ T cells [78,79,80,81]. It becomes aberrantly activated mostly due to increased phosphorylation or the loss of negative regulators [82]. Constitutive activation of STAT5 has been shown to be a direct leukemia driver [83]. As with STAT3, STAT5 can rarely become activated through mutation in a STAT5 isoform itself. STAT5 mutations are commonly found in human hematologic cancers, such as T cell prolymphocytic leukemia, B-cell and T cell acute lymphoblastic leukemia, and γδ T cell-derived lymphomas [5,84,85,86].

STAT5 is expressed in a wide array of mammalian tissues [87]. It was first identified as a “mammary gland factor”, a transcription factor that mediates the effects of prolactin on mammary epithelial cells. It is critical for the growth and differentiation of alveolar progenitors as well as the survival of secretory mammary epithelial cells during normal mammary gland development [88]. Aberrant activation of STAT5 is commonly found in breast cancer, though it is generally associated with more differentiated hormone-responsive tumors, reflecting its physiologic role [89]. It promotes cell survival and instigates breast tumor formation, drug resistance, and metastatic capabilities of breast cancers [90].

In some tumor types, such as acute leukemias and breast cancer, either STAT3, STAT5, or both can be found to be activated inappropriately [91,92]. Although the canonical binding sites for STAT3 and STAT5 appear identical in isolated DNA, their transcriptional and biological effects are distinct [91,92]. For example, STAT3 and STAT5 have antagonistic effects in regulating the transcriptional modulator BCL6, which is a master transcription factor in the regulation and proliferation of B cells and T follicular helper cells [93]. While STAT5 and STAT3 can compete for binding sites to regulate BCL6 expression and lead to opposite effects, STAT5-mediated repression of BCL6 is usually dominant over STAT3-mediated induction because it can displace STAT3 from regulatory regions to which it binds [92,94]. Reflecting these transcriptional effects, co-activation of STAT3 and STAT5 in breast cancer is associated with the more differentiated phenotype of breast cancers than with STAT3 activation alone [91]. This finding may have implications for the effects of inhibitors of specific STATs in cancers in which more than one STAT is activated inappropriately.

**Table 3 cancers-16-01387-t003:** Common direct STAT5 target genes.

Gene	Function	Status	Cell Source	References
*ARNT*	Protein sumoylation	Downregulated	Mouse proB cells	[84]
*BCL2*	Anti-apoptosis	Upregulated	Human T cells	[95]
*BCL2L1*	Apoptosis	Upregulated	Mouse proB cells	[84]
*BCLXL*	Anti-apoptosis	Upregulated	Human T cells	[95]
*C3AR1*	Chemotaxis	Upregulated	Murine proB cells	[78]
*CISH*	STAT inhibitor	Upregulated	Human T cells	[95]
*DUSP1*	Anti-inflammation	Upregulated	Murine proB cells	[78]
*DUSP5*	Anti-proliferation	No change	Human T cells	[95]
*GTF2H5*	DNA repair	Downregulated	Human T cells	[95]
*MBP*	Inflammation	No change	Human T cells	[95]
*MYC*	Proliferation	Upregulated	Murine proB cells	[78]
*OSM1*	Metabolic process	Upregulated	Human T cells	[95]
*PIM1*	Proliferation, survival	Upregulated	Human T cells	[95]
*PIM2*	Cell survival	Upregulated	Murine proB cells	[78]
*RO60*	Sperm antigen	Downregulated	Mouse proB cells	[84]
*RK*	Proliferation	Upregulated	Murine proB cells	[78]
*SERPINA3G*	Proliferation	Upregulated	Murine proB cells	[78]
*SGK1*	Proliferation	Downregulated	Human T cells	[95]
*SLC22A5*	Carnitine uptake	Downregulated	Human T cells	[95]
*SOCS1*	Apoptosis	Upregulated	Murine proB cells	[78]
*SOCS2*	inflammation	Upregulated	Human T cells	[95]
*SRP9*	RNA binding	Upregulated	Mouse proB cells	[84]
*TNFRSf13B*	B cell homeostasis	Upregulated	Murine proB cells	[78]

### 2.3. Other STATs

Although STAT3 and STAT5 are the STAT family members most widely associated with cancer pathogenesis, other STATs can become activated inappropriately in cancer cells and play important biological roles. For example, STAT1 can be activated by various ligands, including IFN-α, IFN-γ, EGF, the platelet-derived growth factor, and IL-6. It has a key role in regulating genes that modulate cell survival, viability, and pathogen response. Reflecting its key role as a mediator of interferon signaling, germline mutations in STAT1 are associated with immunodeficiency [96]. STAT1 mainly acts as an inhibitor of cancer as its expression is associated with a better prognosis [97]. Its activation increases the production of reactive oxygen species and, thus, oxidative stress to selectively sensitize cancer cells in breast cancer [98]. However, reflecting the context-dependency of all STATs, STAT1 was also found to act as a cancer promoter in a mouse model of leukemia [99].

STAT2 is principally activated by type I IFNs [100], and it can form a complex with STAT1 to mediate innate antiviral activity. Mutations in this gene result in immunodeficiency [101]. It can either inhibit or promote tumorigenesis depending on the unique environment presented by each type of cancer [102]. STAT2-mediated initial IFN-I response drives the expression of antitumor IFN-stimulated gene factors that are pivotal in dendritic cell maturation, generation of killer CD8+ T cells, and recruitment of immune cells to the tumor site to restrict tumor growth and metastasis. In contrast, STAT2-sustained late IFN-I signaling promotes the expression of pro-inflammatory mediators and genes involved in chemoresistance and immunosuppression that confer tumor cell survival and disease progression [102]. Overexpression of STAT2 has been associated with outcome changes in human skin cancers, head and neck, kidney, lung, ovary, and endometrium [103,104,105,106].

STAT4 can be activated by cytokines like IFN-α, IFN-β, IL-12, and IL-23 [107,108]. It is required for the maturation of T cells and IFN-γ production [109]. Overexpression of STAT4 can be associated with either better or worse outcomes in cancers, depending on the type of cancer. In epithelial ovarian cancer, activated STAT4 is overexpressed and promotes cancer metastasis via tumor-derived Wnt7a-induced activation of cancer-associated fibroblasts [110]. In head and neck squamous cell carcinoma, however, STAT4 mediates resistance to metastasis, and activation of STAT4 could potentially mitigate lymphatic metastasis [111]. Similarly, high expression of STAT4 predicted better clinical outcomes in gastric cancers [112].

STAT6 can be activated by growth factors and cytokines such as IL-4 and IL-13 [113]. It has been reported to be highly expressed in several types of cancer, including breast, pancreatic, prostate, and colorectal cancer [114]. In colon cancers, STAT6 regulates mechanisms that promote the proliferation, survival, invasion, and metastasis [114,115,116]. In breast cancer, targeting the STAT6 pathway inhibits protumorigenic and prometastatic activities induced by tumor-associated macrophages in both in vitro and in vivo models [117].

## 3. Targeting STATs for Cancer Therapy

It is clear that inappropriately activated STATs directly drive the malignant phenotype of cancer cells. However, the key issue in developing cancer therapeutics is to have an acceptable therapeutic index, the ability to kill cancer cells without harming normal cells [118]. Given the central role that STATs play in so many central physiologic processes, there was a concern that STAT inhibitors might have unacceptable toxicities [12]. However, evidence form a number of areas has suggested that STAT inhibition can be well-tolerated, especially for relatively brief intervals needed for cancer therapy [119,120,121].

One piece of evidence of the tolerability of STAT3 inhibition came from the discovery that the inherited hyper-IgE syndrome (sometimes referred to as Job’s syndrome) is caused by one of several mutations in STAT3 [122]. Not only do these mutations inactivate the transcriptional function of the affected allele, but given the requirement that STAT3 form a dimer to mediate transcriptional regulation, these mutant forms of STAT3 act in a dominant inhibitory manner [123]. Consequently, individuals with these inherited mutations have greatly reduced STAT3 transcriptional activity from the time of conception [124]. While various inflammatory and immune-deficient effects characterize hyper-IgE syndrome, these individuals develop normally [125]. Thus, severe attenuation of STAT3 function in every cell in the body throughout development and beyond can be tolerated. This finding, coupled with evidence from animal models and the use of pharmacologic inhibitors of STAT function, has supported the concept that inhibition of STAT transcriptional function could be a targeted form of cancer therapy that would be well tolerated [126,127]. Furthermore, since STATs sit at a convergence point of multiple upstream kinases, targeting STATs holds the potential to be efficacious in a broad array of cancers.

One of the most significant recent advances in cancer therapy has come through the development of kinase inhibitors, which can target mutated, over-expressed, or even normally expressed wildtype kinases to which the cancer cell has become dependent or “addicted” [128,129]. While these agents can be enormously effective in treating patients even with advanced disease, a near-universal shortcoming of these drugs is the emergence of resistance [130,131]. Resistance to targeted anti-cancer therapies like kinase inhibitors often arises from the activation of another tyrosine kinase or a parallel signaling pathway. For example, one mechanism for the emergence of resistance to EGF receptor tyrosine kinase inhibitors in patients with non-small-cell lung cancer is the activation of the MET receptor tyrosine kinase [132]. These parallel signaling pathways often still converge on the same small number of oncogenic transcription factors, like STATs [133,134]. Therefore, in addition to the direct therapeutic benefits from targeting STAT proteins, their position as convergence points for multiple signaling pathways holds the promise that it will be more difficult for resistance to develop to these agents and that they may be particularly useful in combination with other targeted therapies [135].

Three main approaches have been pursued in considering how best to therapeutically target STATs, which will be discussed in more detail. The first is to target STAT proteins directly. This has often been challenging, as transcription factors are generally considered structurally difficult to inhibit with small molecules [136]. This reflects the fact that, as opposed to kinases, which have discrete ATP-binding domains in which a small organic molecule can be designed to bind, transcription factors have large, flat surfaces that allow them to interact with DNA or other proteins [137]. Nonetheless, STATs possess structural elements like SH2 domains that have been used for direct targeting [138,139]. The second approach is to use screening strategies, such as with chemical biology or computational methods, to identify compounds that specifically inhibit STAT-dependent transcription [140,141]. The third approach is to target steps in STAT activation that are directly upstream of STAT phosphorylation [142]. Most therapeutic development in this area has been directed at STAT3, though, as noted, other STATs may also play pathogenic roles in a variety of cancers (Figure 2).

## 4. Strategies to Directly Target STATs

### 4.1. Direct STAT Binding Molecules

Almost all compounds designed to target STAT3 in clinical development currently do so by targeting the SH2 domain of the protein [138] (Table 4). Since the STAT3 SH2 domain participates in dimerization with another STAT3 protein to form a STAT3 homodimer or with a STAT1 protein to form a STAT1-STAT3 heterodimer, peptides that can bind to the STAT3 SH2 domain are relatively easy to design [143]. However, small peptides, particularly ones that need to be phosphorylated or otherwise have a strong negative charge like one that would bind to the STAT3 SH2 domain, generally have unfavorable pharmacologic characteristics, including rapid degradation and limited intracellular penetration [144]. Nonetheless, non-peptide molecules derived from this approach have been made and have shown some efficacy in model systems [139,145]. One potential shortcoming of this approach is that SH2 domains have some latitude in the sequences they can bind to, and different SH2 domains can bind to the same tyrosine–phosphorylated sequence [146]. Consequently, it is difficult to develop molecules that can bind to the STAT3 SH2 domain with a high degree of specificity. Nonetheless, a molecule that bound with high specificity to the STAT3 SH2 domain would be appealing, as it would inhibit the recruitment of STAT3 to activated receptor–kinase complexes and block the activating dimerization of STAT3.

Among the small molecules, BP-1-102 and its two analogs were designed as direct STAT3 inhibitors with reasonable in vivo tumor-inhibiting activity by binding specifically to the STAT3 SH2 domain [161]. N4 is another small molecule that has potent antitumor bioactivity. It directly binds to the STAT3 SH2 domain and thereby inhibits the STAT3 dimerization and STAT3-NF-κB cross-talk [139]. Betulinic acid is a plant-derived compound that has been found to induce cell apoptosis in cervical cancer. It has a very high affinity to STAT3 SH2 and thus also inhibits STAT3 phosphorylation, as would be predicted by this mechanism of action [162,163].

### 4.2. STAT Degraders

Although direct STAT-binding molecules such as small-molecule STAT3 inhibitors are promising therapeutic strategies, they often have selectivity problems and show very limited clinical activity [133]. In recent years, targeted protein degradation (TPD) strategies that leverage a cell’s endogenous protein destruction machinery to remove specific disease-associated proteins have emerged as promising approaches. Proteolysis–targeting chimeras (PROTAC) and molecular glues are two examples of TPD strategies [164].

PROTACs, also known as bivalent chemical protein degraders, are heterobifunctional molecules that degrade specific endogenous proteins through the E3 ubiquitin ligase pathway [165,166,167]. PROTACs have the potential to be better tolerated than traditional small molecule inhibitors as PROTACs exert their effects through a repeated and iterative mode of action to induce target protein degradation rather than competing with active sites like traditional small molecule inhibitors [168]. Several PROTAC degraders have been developed to target STATs [167]. Many of these molecules employ a STAT3-binding component that binds to the STAT3 SH2 domain. For example, SD-36 consists of an analog of the CRBN ligand lenalidomide, a linker, and the SH2-targeting STAT3 inhibitor SI-109 [169]. It achieves tumor regression in multiple xenograft models, including acute myeloid leukemia, anaplastic large-cell lymphoma, and glioma [121,170]. SD-91 is another STAT3 degrader that is capable of achieving complete and long-lasting tumor regression in xenograft models of megakaryoblastic leukemia [171].

Recently, a phase 1 clinical trial was initiated using the STAT3 degrader KT-333 in a variety of hematologic cancers and solid tumors [172]. In addition to evidence of some clinical activity, robust pharmacodynamic data have been obtained that show significant downregulation of STAT3 protein expression, reflecting an on-target effect [172]. Furthermore, the canonical STAT3 target gene *SOCS3* was downregulated in treated patients, as were the STAT3-dependent acute-phase inflammatory markers C-reactive protein (CRP) and serum amyloid A (SAA) protein [172].

Molecular glues are small molecules that can induce protein–protein interactions between a ubiquitin ligase and a target protein, which leads to protein ubiquitination and subsequent proteasome-based degradation [173]. Molecular glues may have advantages over heterobifunctional PROTACs, including favorable physicochemical properties [174]. Molecular glues targeting STAT3 are still in developmental phases, though this technology holds great potential to be exploited [175].

## 5. Targeting Upstream Kinases: STAT Phosphorylation as a Biomarker for On-Target Effects

Since STAT activation in the tumor microenvironment is often driven, at least in part, by the presence of cytokines that can activate STATs, targeting these effects has been an appealing strategy. However, for a number of reasons, targeting upstream steps in these pathways has been therapeutically useful only in a relatively small number of circumstances.

IL-6 can activate STAT3, and IL-6 is often elevated systemically and in the tumor microenvironment in patients with cancer [176]. Furthermore, IL-6 levels can be a negative prognostic indicator in cancer patients. Since antibody-based therapeutics that either bind directly to IL-6 or block the effect of IL-6 by binding to its receptor are already in use for inflammatory and rheumatologic disorders [177,178], several clinical trials have tested this approach (NCT04333706, NCT04940299, and NCT02644967). However, by and large, these strategies have shown little clinical benefit. There are several reasons for this. The first is that while IL-6 can lead to the activation of STAT3 in the tumor microenvironment, many other cytokines and growth factors can do so as well. Multiple STAT3-activating cytokines can also be detected in the conditioned medium of primary cancer cells or cancer cell lines grown in vitro [133]. Thus, blocking a single cytokine has limited therapeutic efficacy. Even where successful, the rapid emergence of resistance is likely to occur. While it is true that elevated levels of IL-6 may be associated with a worse prognosis in cancer patients [179], this can also reflect the production of IL-6 in response to the physiologic effects of having advanced cancer, whereby the acute phase response is induced [180]. In that setting, merely inhibiting IL-6 is unlikely to have significant therapeutic benefits.

A second approach is to target kinases downstream of cytokines, such as JAK inhibitors [181]. While JAK inhibitors can decrease STAT3 tyrosine phosphorylation in tumor cells, this approach suffers from two major shortcomings. The first reflects the fact that cytokine receptors are generally associated with two different JAK family members. Receptors that mediate signaling via the gp130 receptor chain (which transduces signals from IL-6, LIF, oncostatin M, IL-13, CNTF, and others) associates with three different JAK family members (JAK1, JAK2, and TYK2) [182]. Therefore, to effectively suppress signaling through these receptors, kinase inhibitors would need to suppress almost all cytokine signaling in a patient [183]. Such an approach would lead to unacceptable toxicity [184].

The second limitation of JAK inhibitors, even more selective ones, reflects the fact that the immune response to cancer is often a critical component of the therapeutic response. A key mediator of this effect, both in making tumor cells more visible to immune cells and activating immune cell function, are IFNs [185,186]. JAK inhibitors will generally suppress IFN signaling (which also uses JAK1, JAK2, and TYK2) and thus suppress critical immune effects [6]. This has also limited the clinical applicability of JAK inhibitors in cancer therapy.

Despite these limitations, as outlined below, there are some clinical situations in which inhibition of upstream kinases is an effective approach to suppressing pathogenic STAT signaling. Furthermore, measuring STAT phosphorylation and target gene expression can also be an effective pharmacodynamic marker for the therapeutic activity of kinase inhibitors.

### 5.1. Inhibiting STAT5 in Chronic Myeloid Leukemia

It has been known since the 1970s that chronic myeloid leukemia (CML) is almost universally associated with a translocation of chromosomes 9 and 22, which leads to a fusion oncoprotein, BCR-ABL1 [187]. This fusion tyrosine kinase can phosphorylate STAT5 directly and also lead to the activation of JAK2 in both CML and some forms of acute lymphoid leukemia (ALL) [188,189,190,191]. As noted, cellular and enzymatic analyses suggest that STAT5 is phosphorylated by BCR-ABL1 directly and that STAT5 is indispensable for the initial transformation of leukemia [192]. In contrast, initial myeloid transformation and leukemia maintenance were independent of JAK2. These results suggest that it is more effective to target STAT5 rather than JAK2 to treat BCR-ABL+ diseases [192]. Inhibiting BCR-ABL1 with tyrosine kinase inhibitors like imatinib has revolutionized the treatment of CML, and suppresses STAT5 phosphorylation [193]. Resistance to first-generation BCR-ABL1 kinase inhibitors like imatinib can occur. However, compounds that inhibit STAT5 activation through kinase-independent mechanisms, like pimozide, still have activity in this setting [194]. Of note, combined inhibition of STAT5 with pimozide may synergize with JAK2 inhibition in models of other myeloproliferative neoplasms [195].

### 5.2. Targeting Kinases Upstream of STATs in AML

Given the importance of STATs in mediating the oncogenic effects of upstream signaling pathways, monitoring the phosphorylation of STATs in clinical samples may prove to be an important method to ensure on-target effects by kinase inhibitors. This approach may also be useful as an early means to detect resistance to these therapies. This type of strategy was exemplified in a recent clinical trial for patients with relapsed or refractory acute myeloid leukemia (AML). Mesenchymal epithelial transition (MET) is an oncogenic receptor tyrosine kinase that is upregulated or overly activated in many cancers, including hematopoietic cancers like AML [196]. Autocrine production of HGF can activate MET by binding to its ligand FGF receptor (FGFR) and eventually lead to myeloblast growth and survival in acute myeloid leukemia [197,198]. MET inhibitors such as merestinib exhibit activity in AML preclinical studies, but HGF upregulation by the FGFR pathway is a common mechanism of resistance [199]. Based on this background and in vitro studies that suggested efficacy from targeting both the MET and FGF receptor pathways simultaneously in AML, a phase 1 clinical trial using the rational combination of the MET inhibitor merestinib and the FGFR inhibitor LY2874455 was conducted in patients with relapsed or refractory acute myeloid leukemia [200]. Although the study was restricted by limitations in the supply of the drugs, one key finding was that, at least in some subjects, phosphorylation of STAT3 and/or STAT5 correlated closely with clinical response. This suggests that readily available techniques, such as flow cytometry, could be useful in monitoring and regulating the dose of kinase inhibitors in certain settings. This study also found that in the setting of progressive disease, genes that were upregulated could activate STAT3 and/or STAT5 through alternate mechanisms. This finding again emphasizes the point that STATs, which sit at a convergence point for multiple upstream kinase pathways, including MET [201], may be a more effective target than upstream kinases alone [202].

## 6. Novel Ways to Identify STAT Transcriptional Inhibitors

Searching for structural elements in a protein through which therapeutic inhibitors can be developed is a standard approach to drug development [203]. However, transcription factors are not optimal proteins for this strategy. This reflects their relative lack of surfaces to which small organic compounds can bind, which leads to functional inhibition [204]. As noted, the potentially targetable motifs that they do possess, such as the SH2 domain, may not confer optimal specificity for drug development [144]. Therefore, alternate approaches to developing compounds that block STAT-dependent transcriptional activity are appealing. In addition to leading to novel therapeutic compounds, this strategy can help to uncover unappreciated aspects of STAT-dependent gene regulation. In addition, these types of approaches may also be able to identify compounds that can enhance STAT-dependent transcriptional activity, which may be useful in certain applications. Finally, these broad-based and unbiased strategies can also be applied to other transcription factors.

Given that much is known about molecular aspects of STAT transcriptional regulation, two screening strategies that have leveraged this information have been particularly fruitful, which will be described below. One is dependent on screening compounds that specifically alter the transcriptional activity of a specific STAT family member [205]. The second uses a strategy that leverages large databases of gene expression changes induced by drugs to infer compounds that can specifically modulate STATs [206]. Both of these approaches have led to the identification of STAT inhibitors that have been introduced into therapeutic clinical trials for cancer therapy.

### 6.1. Chemical Biology Approaches

The general motif for the binding of STATs to gene regulatory regions has been well-defined. For most STATs, the consensus genomic cognate binding site conforms to the general nine-base pair sequence TTCNNNGAA [7]. One can, therefore, generate heterologous reporter genes in which one or more STAT consensus sequences are placed upstream of a reporter gene, such as luciferase [207]. These constructs can then be introduced into cells with undetectable basal STAT activation, in which a specific STAT can be activated with an appropriate cytokine [207]. Such a system can then be tested on large chemical libraries to identify molecules that can inhibit (or enhance) the STAT-dependent luciferase activity (Figure 3). To ensure the specificity of any molecules identified, the chemical library would also be screened against parallel systems in which other transcription factors drove the expression of the reporter gene.

This approach has identified a number of compounds that act through a variety of mechanisms. One drug identified in this manner as an inhibitor of STAT3 transcriptional function is nifuroxazide [205]. Nifuroxazide appears to act mainly through inhibition of JAK kinase activity and showed activity against STAT3-dependent multiple myeloma cell lines and primary cells. As expected from this mechanism of action, it was also able to overcome the pro-survival effects afforded by co-culture with bone marrow stromal cells [205].

While nifuroxazide acts through a fairly conventional mechanism, other compounds identified through this approach reflected novel mechanisms. For example, the diphenylbutylpiperidine anti-psychotic drug pimozide, which is used to treat the tics associated with Tourette syndrome, was identified to inhibit both STAT5 and STAT3 transcriptional activity [194]. In contrast to nifuroxazide, pimozide is not a direct inhibitor of any JAK family member or BCR-ABL1 [195]. Although the exact mechanism of action is still being elucidated, pimozide appears to act by enhancing the activity of negative regulators of STAT signaling [208]. By virtue of having this kinase-independent effect, pimozide has shown efficacy in leukemic models driven by BCR-ABL1, mutated JAK2, and mutated FLT3 [209].

Both nifuroxazide and pimozide decrease the activating tyrosine phosphorylation of STATs. Another compound identified through this chemical biology approach, pyrimethamine, decreases STAT3-dependent transcription but does not appear to significantly decrease STAT3 phosphorylation, nuclear localization, or DNA binding [210,211,212,213]. Pyrimethamine likely inhibits the interaction between STAT3 and other proteins necessary for transcriptional activation. This discovery through an unbiased approach may reveal other strategies for targeting STAT3-dependent gene expression [206].

Since pyrimethamine is already used as an anti-parasitic agent treating diseases such as toxoplasmosis and malaria, abundant human pharmacokinetic and safety data about this drug are available [214]. Pyrimethamine inhibits STAT3-dependent transcription at low micromolar concentrations, which is known to be safely achieved in humans for months at a time [214]. This allowed the planning of a clinical trial of pyrimethamine in patients with chronic lymphocytic leukemia (CLL), the most common form of leukemia in much of the world [150]. CLL is characterized by the transcriptional activation of STAT3 in almost every patient, though it is through a non-canonical phosphorylation event [215,216]. In this clinical trial, patients with CLL who had progressed despite multiple lines of standard therapy were treated with single-agent pyrimethamine [150]. Although the mean plasma levels of pyrimethamine (6.17 μM) were at the lower end of the concentration needed to inhibit STAT3 transcriptional activity, half of the patients achieved stable disease [150]. To understand whether this drug was inhibiting STAT3 transcriptional activity in the leukemia cells of these patients, cells were isolated from their blood, and RT-PCR analyzed the expression of a panel of STAT3-dependent genes. Suppression of STAT3-dependent gene expression was seen in 50% of the patients. Notably, at the time of disease progression, increased expression of the STAT3 signature genes was generally observed, suggesting that this drug was working through an on-target mechanism [150]. This type of integrated analysis of clinical efficacy with pharmacodynamic measurements will be important in the further therapeutic development of STAT inhibitors.

While it is therapeutically useful to identify inhibitors of oncogenic STAT family members that drive cancer pathogenesis, these types of chemical biology approaches can also be useful in identifying putative activators of transcription factors. As noted earlier, STAT1 mediates the effects of IFNs and is a key component of the innate immune system response to viral infections and other pathogens [217]. Activated STAT1 can promote cell cycle arrest, apoptosis, differentiation, and enhanced immune recognition [6]. It can also mediate anti-angiogenic effects [218]. In fact, inhibition of STAT1 may mediate the immunosuppressive effects of drugs like fludarabine [219]. A similar chemical biology approach has been used to identify activators of STAT1-dependent transcription [220]. This strategy identified 2-(1,8-naphthyridin-2-yl)phenol (2-NP) as a compound that could enhance STAT1-dependent gene expression. It extends the duration of IFN-γ-induced STAT1 tyrosine phosphorylation and, in this way, may amplify signals through this transcription factor. The 2-NP enhanced the ability of IFN-γ to decrease the proliferation of human breast cancer and sarcoma cell lines. Indicating that this effect was mechanism-specific, cells that lacked STAT1 were unaffected by either IFN-γ or 2-NP [220]. It remains to be seen whether enhancing STAT1 activity will be a worthwhile addition to anti-cancer therapy.

### 6.2. Computational Approaches Leveraging Transcriptional Signatures

In addition to chemical biology approaches, other open-ended and unbiased strategies have also been useful in identifying inhibitors of STAT-dependent gene expression. The target genes directly modulated by STAT transcription factors have been increasingly defined, particularly those regulated by STAT3 in cancer cells [221]. At the same time, publicly accessible data sets, such as the Connectivity Map, have become available, providing large amounts of data on how a wide variety of chemical compounds alter gene expression in a variety of cell types [222]. STAT3 target genes can then be ordered from the ones most highly induced by STAT3 to those most highly repressed by STAT3. Data sets such as the Connectivity Map can then be employed to identify compounds that are associated with the exact opposite effect—decreasing expression of STAT3-induced genes and increasing expression of STAT3-repressed genes [223]. The hypothesis is that the compounds identified by this strategy would likely be inhibitors of STAT3 transcriptional function (Figure 4).

Using this approach, it was found that atovaquone induced gene expression changes that were highly anti-correlated with a STAT3 gene expression signature [222]. From this purely computational or in silico method, subsequent experiments revealed that atovaquone did, in fact, decrease STAT3 tyrosine phosphorylation. Atovaquone is not a kinase inhibitor but rather decreased signaling through the gp130 (CD130) receptor chain used by IL-6 and many other family members to cause the phosphorylation of STAT3. Atovaquone decreased the survival of STAT3-dependent cell lines and primary leukemic cells [153]. Atovaquone is widely used in oncology to prevent the development of *Pneumocystis* pneumonia in immunosuppressed patients [224]. From the pharmacokinetic data available, it is clear that levels of atovaquone sufficient to suppress STAT3 phosphorylation were readily and safely achieved in patients [206,225]. In fact, serum from patients taking atovaquone can be shown to have anti-leukemic effects compared to serum from patients taking other drugs for prophylaxis of *Pneumocystis pneumonia* [226]. Retrospective data also provided evidence that after allogeneic hematopoietic stem cell transplantation for AML, patients receiving more atovaquone were less likely to experience disease relapse. Consequently, clinical trials testing the anti-cancer effects of atovaquone are currently in progress for STAT3-driven cancers, including ovarian cancer (ClinicalTrials.gov NCT05998135) and AML (ClinicalTrials.gov NCT03568994).

## 7. Conclusions and Future Directions

Oncogenic transcription factors such as STAT3 and STAT5, which sit at convergence points of multiple upstream pathways, are appealing targets for cancer therapy. Although they are not commonly mutated themselves, they mediate the oncogenic effects of many diverse upstream oncogenic events. Since so many pathways converge on a relatively small number of proteins that regulate the genes mediating malignant cellular behavior, inhibition of STATs holds the promise of low rates of resistance arising from the activation of alternate or parallel pathways [12]. Finally, the fact that loss of STAT activity is tolerated well in healthy cells suggests that targeting STAT transcription factors will have a high therapeutic index.

While targeting transcription factors can be challenging, the increasing success in achieving this goal makes the oft-used term “undruggable” inappropriate. Through a combination of a direct targeting approach and chemical biology and computational strategies, novel ways to target this pathway are becoming apparent. The agents uncovered from these approaches may be useful therapeutic agents themselves. Perhaps more importantly, they may lead to unappreciated mechanisms that can be exploited further with sophisticated medicinal chemistry approaches.

One other aspect of targeting STATs for cancer therapy should be noted. As with most cancer treatments, single-agent therapy may not be fully successful [227]. While an oncogenic transcription factor such as STAT3 or STAT5 may regulate genes that underlie a cancer phenotype, interrupting these pathways may not be sufficient to kill established tumors. However, inhibiting oncogenic STATs in cancer may set the stage for synthetic lethal combination strategies [127]. For example, many target genes of STATs encode pro-survival proteins such as BCL2, BCL-XL, MCL1, and survivin [228]. By virtue of inhibiting the expression of these proteins, STAT inhibitors may allow synergy with cytotoxic drugs or radiation.

In addition, STAT target genes can affect DNA repair mechanisms, suggesting that combinations of STAT inhibitors with PARP inhibitors or telomerase inhibitors may show enhanced efficacy [229].

Furthermore, STAT3 mediates the physiologic acute phase response. One component of this response involves protecting healthy cells from being killed by infiltrating immune cells in the setting of tissue injury, inflammation, and infection [230,231]. Increased STAT3 activity in a tumor likewise protects a tumor from immune-based destruction. Therefore, combinations of STAT inhibitors and immune-activating therapies, both immune checkpoint inhibitors and engineered cellular therapies, may be a promising approach [232].

Given these ongoing discoveries, there is a high likelihood that further advances in targeting STATs and other oncogenic transcription factors will be a major component of a new generation of cancer therapies that display increased efficacy and decreased toxicity in the coming years.

## Figures and Tables

**Figure 1 cancers-16-01387-f001:**
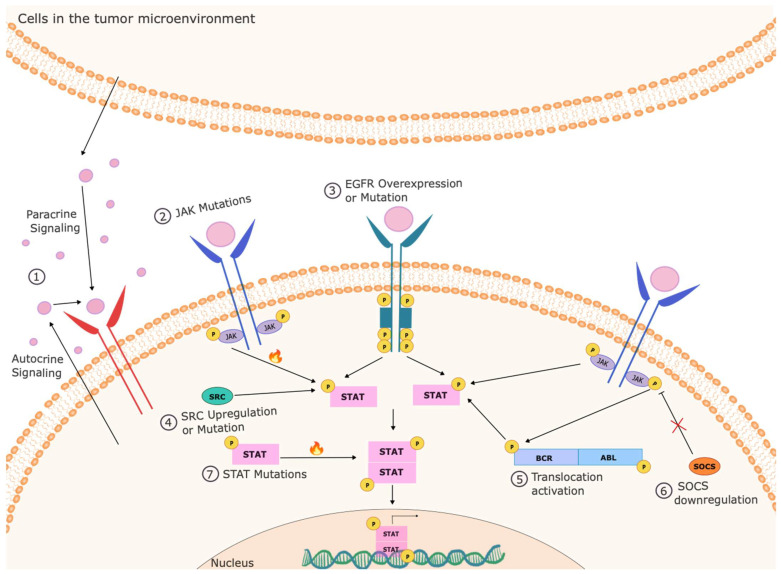
Mechanisms of inappropriate STAT activation in cancer cells. STATs can be activated constitutively in cancer cells through a variety of mechanisms. This can include autocrine or paracrine production of cytokines that can activate these pathways, activation of upstream kinases through mutation (such as in JAKs), overexpression (such as EGFR), inappropriate activation (such as SRC), or activating translocations (such as in BCR-ABL), and loss of negative regulators (such as SOCS3). Rarely, the STATs can be activated by mutation within the STAT itself.

**Figure 2 cancers-16-01387-f002:**
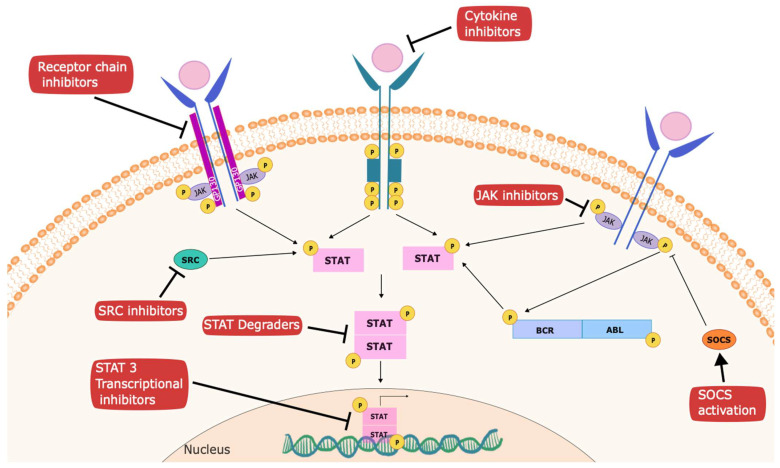
Strategies to inhibit STATs for cancer therapy. Several targets have been established to mediate anti-cancer effects within the STAT signaling pathway. These include inhibitors of upstream kinases, such as JAKs, SRC, and receptor tyrosine kinases; inhibitors of receptor chains like GP130 (such as atovaquone); activators of negative regulators, such as SOCS family members; degraders of STATs; and agents that interfere with recruitment of transcriptional complexes, such as pyrimethamine.

**Figure 3 cancers-16-01387-f003:**
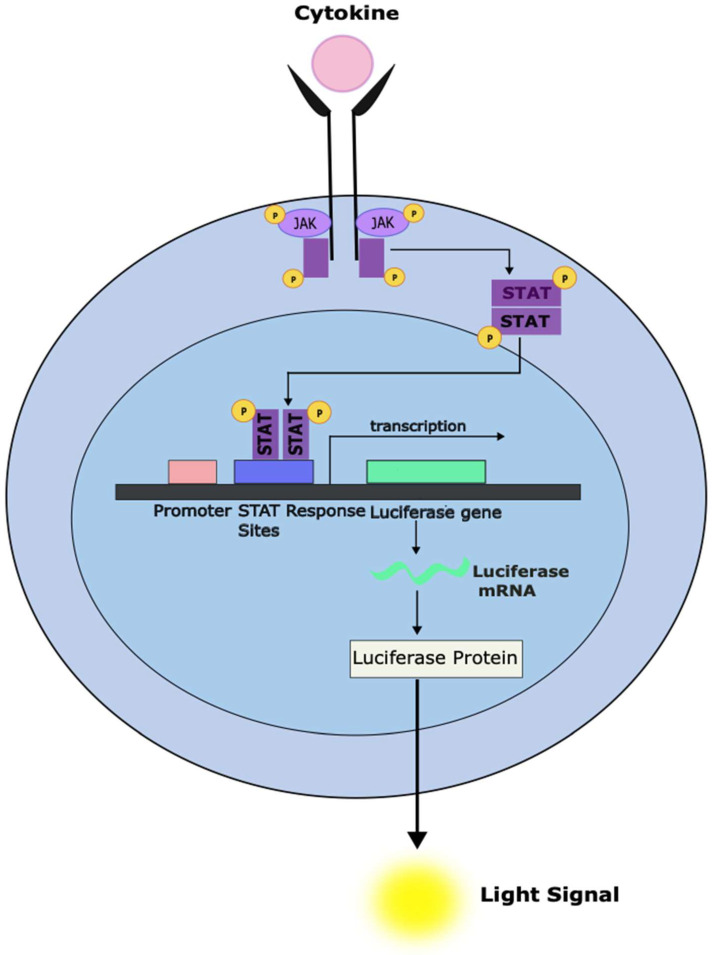
Development of a cell-based system to screen compounds for the ability to inhibit STAT-dependent transcriptional activity. The stable introduction of a luciferase reporter gene under the control of a STAT-responsive promoter can enable a system to rapidly screen large numbers of compounds for the ability to inhibit STAT-dependent gene transcription (as measured using luminometry). Cellular systems can be designed to only respond to a specific STAT family member. It is also essential to counter-screen against similar systems under the control of unrelated transcription factors to exclude compounds that appear active in this assay but act through non-specific effects like general cytotoxicity.

**Figure 4 cancers-16-01387-f004:**
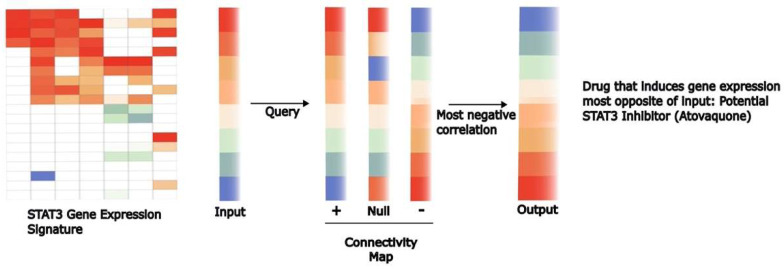
Gene expression signatures coupled with databases of drug-induced changes in gene expression can be used to identify potential transcriptional inhibitors. General signatures of gene expression changes mediated by specific transcription factors have been defined. The availability of large databases, such as the Connectivity Map, provides an opportunity to query a large number of drugs to identify those that induce gene expression changes most negatively correlated with the gene signature being queried. “Hits” from this computational approach hold the potential to be inhibitors of the specific pathway. Through this approach, the anti-parasitic drug atovaquone was identified as a therapeutically accessible inhibitor of STAT3-dependent transcription.

**Table 1 cancers-16-01387-t001:** Overview of preclinical models in the field of cancer.

Methods	Preclinical Models	Expense	Advantage	Disadvantage
In vitro	Immortalized cell lines	Low	Can be easily maintained and expanded	Genetic instability and the occurrence of clonal selection
	Primary 2D cultures	Low	High take rate, amenable to genetic manipulation	Inability to reflect the histological nature
	3D organoids	Moderate	Useful to study the interactions between different cell populations	Do not fully reproduce the complexity, lower sensitivity of cells
In vivo	*Drosophila melanogaster*	Low	Gives insight into asymmetric division. Genetic similarity with humans	Rudimentary hematopoietic systems and different lymphatic system
	Zebrafish	Moderate	Rapid development, chemical screening, amenable genetics, and fitness for in vivo imaging	Difficulty in the examination of fixed tissue, low tumor incidence
	Patient-derived xenografts	Expensive	Conservation of a stromal compartment, tumor tissue expansion	Lack of a functional immune environment in PDX, prolonged time needed for model establishment and expansion compared to organoids
	Carcinogen-induced mouse models	Expensive	Suitability to study effects of carcinogenic and genetic factors in tumorigenesis	Extended time needed to develop full-fledged carcinomas
	Genetically engineered mouse models	Expensive	Closely recapitulate the heterogeneous landscape of genomic alterations in human primary tumors	Only a fraction of mutations drive tumorigenesis by affecting oncogenes or tumor suppressor genes
	Pig cancer models	Expensive	Efficiently represent the progression and development of cancer in humans	Biosafety issues, larger housing requirements, longer generation intervals, and fewer genomic tools

**Table 4 cancers-16-01387-t004:** Inhibitors of STAT3 or STAT5 in clinical development.

Type	Agent	Target	Cancer Type	ClinicalTrial.gov Identifier	Phase	References
Small molecules	Silibinin	STAT3	Endometrial carcinoma		Preclinical	[147]
	SD-36	STAT3	Acute myeloid leukemia and anaplastic large-cell lymphoma		Preclinical	[121]
	BP-1-102	STAT3	Acute lymphoblastic leukemia		Preclinical	[148]
	LLL12	STAT3	Ovarian cancer		Preclinical	[149]
	Pyrimethamine	STAT3	Chronic lymphocytic leukemia	NCT01066663	Phase 1/2	[150]
	OPB-51602	STAT3	Nasopharyngeal carcinoma	NCT01184807	Phase 1	[151]
	N4	STAT3	Pancreatic cancer		Preclinical	[139]
	Atovaquone	STAT3	Non-small cell lung cancer	NCT02628080	Phase 1	[152]
		STAT3	Acute myeloid leukemia		Preclinical	[153]
	Trichothecin	STAT3	Colorectal Cancer		Preclinical	[154]
	SDL-1	STAT3	Gastric cancer		Preclinical	[155]
	AK-2292	STAT5	Chronic myeloid leukemia		Preclinical	[156]
Oligonucleotides	Danvatirsen	STAT3	Diffuse large B cell lymphoma	NCT03527147	Phase 1	[157]
		STAT3	Myelodysplastic syndromes, acute myeloid leukemia	NCT05986240	Phase 1	[158]
	Double-stranded minicircles	STAT3	Triple-negative breast cancer		Preclinical	[159]
Peptides	OPB-31121	STAT3	Hepatocellular carcinoma	NCT01406574	Phase 1/2	[157]
	PS-acet.-STAT3 peptide	STAT3	Melanoma		Preclinical	[160]

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
