# Peer review of "Oncogenic STAT Transcription Factors as Targets for Cancer Therapy: Innovative Strategies and Clinical Translation"

_cancers, 2024, doi:10.3390/cancers16071387_

Round 1

Reviewer 1 Report

Comments and Suggestions for Authors

Main advice: The purpose of writing a review article in the natural and medical sciences is not to enumerate previous findings. By determining how reliable those previous findings are physiologically or pathologically, we can rearrange the chaotic frontiers; for example, by organizing and presenting the most universally reliable biological functions from top (i.e., the most universally or pathologically important one) to bottom, by cancer type, or by assay systems (i.e., a mouse model or cell lines they used), we can reduce unnecessary confusion. Constitutive or deregulated expression of a protein of interest in particular cancer, which is an interesting or informative observation in itself, does not immediately prove its causative function. It is always required to identify what kind of 'functional' assays they used. Rearranging and discriminating the various biological findings into a reliable, functional order is the goal of writing a review article through which we can find a glimmer of light in the current confusing situation. What I found particularly confusing was section 2.4. After reading to the end of chapter 2, I was just confused. Despite a lot of expression and function research cited here, my honest impression is that STAT research still seems immature as an academic discipline. If I were a regular reader, I would stop reading at that point. According to the title, the main message exists in/after chapter 3. If so, this reviewer strongly urges the authors to re-write, revise, or if appropriate, remove part of the first two confusing chapters.

Minor (but still important) suggestions

1) Figures 1 and 2 are puzzling to interpret. Different cytokines seem to represent different color balls, but there is no explanation of what they are in the figure legends. Why does only EGFR exist three? Any rationale to have three EGFR there? All fonts in Figures 1 and 2 (and also in Figure 3) are too small to read.

2) All Tables 1, 2, and 3 are structurally disordered.

Comments on the Quality of English Language

If rearranged in a order of functional and pathological importance, their scientifically confusing sentences will be drastically improved.

Author Response

Comments and Suggestions for Authors

Main advice: The purpose of writing a review article in the natural and medical sciences is not to enumerate previous findings. By determining how reliable those previous findings are physiologically or pathologically, we can rearrange the chaotic frontiers; for example, by organizing and presenting the most universally reliable biological functions from top (i.e., the most universally or pathologically important one) to bottom, by cancer type, or by assay systems (i.e., a mouse model or cell lines they used), we can reduce unnecessary confusion. Constitutive or deregulated expression of a protein of interest in particular cancer, which is an interesting or informative observation in itself, does not immediately prove its causative function. It is always required to identify what kind of 'functional' assays they used. Rearranging and discriminating the various biological findings into a reliable, functional order is the goal of writing a review article through which we can find a glimmer of light in the current confusing situation. What I found particularly confusing was section 2.4. After reading to the end of chapter 2, I was just confused. Despite a lot of expression and function research cited here, my honest impression is that STAT research still seems immature as an academic discipline. If I were a regular reader, I would stop reading at that point. According to the title, the main message exists in/after chapter 3. If so, this reviewer strongly urges the authors to re-write, revise, or if appropriate, remove part of the first two confusing chapters. 

Thank you for these comments.  Section 2 addresses the important point of the role of STAT transcription factors in driving cancer pathogenesis.  This background is important to appreciate the therapeutic strategies highlighted in the subsequent sections.  Based on the reviewer’s comments, we have rearranged and rewritten parts of section 2 to allow a more clear and logical flow of ideas.

Minor (but still important) suggestions

1) Figures 1 and 2 are puzzling to interpret. Different cytokines seem to represent different color balls, but there is no explanation of what they are in the figure legends. Why does only EGFR exist three? Any rationale to have three EGFR there? All fonts in Figures 1 and 2 (and also in Figure 3) are too small to read.

These suggestions are very much appreciated.  We have simplified these figures, increased the font size, and tried to clarify the legends to the extent possible.

2) All Tables 1, 2, and 3 are structurally disordered.

We have adjusted the layout of the tables, while trying to conform to the journal requirements. The tables will likely need editorial adjustment to meet the format requirement of the journal.

Comments on the Quality of English Language

If rearranged in a order of functional and pathological importance, their scientifically confusing sentences will be drastically improved.

As noted, we have rearranged section 2 to improve the flow of concepts.

Reviewer 2 Report

Comments and Suggestions for Authors

In the Review Article entitled “Oncogenic STAT Transcription Factors as Targets for Cancer Therapy: Innovative Strategies and Clinical Translation”, Wang and Co-authors reported a critical analysis of currently available data on the role of STAT proteins in different aspects of cancer development, focusing attention on STAT3 and STAT, two members most frequently activated in cancers and able to drive neoplastic development. In this context, the authors also highlighted the potential therapeutic implications. The topic addressed is very interesting and well discussed by the authors and could provide useful information to define new promising approaches based on STAT.

Collectively, the paper is well written and structured in a clear form. Therefore, only a minor revision is required to be accepted.

Minor revision:

-          Chapter 2 begins with a general introduction and then moves on to paragraph 2.2. Paragraph 2.1 is missing. Please, check the subdivision.

-          Please, check the layout of the tables

Author Response

In the Review Article entitled “Oncogenic STAT Transcription Factors as Targets for Cancer Therapy: Innovative Strategies and Clinical Translation”, Wang and Co-authors reported a critical analysis of currently available data on the role of STAT proteins in different aspects of cancer development, focusing attention on STAT3 and STAT, two members most frequently activated in cancers and able to drive neoplastic development. In this context, the authors also highlighted the potential therapeutic implications. The topic addressed is very interesting and well discussed by the authors and could provide useful information to define new promising approaches based on STAT. 

Collectively, the paper is well written and structured in a clear form. Therefore, only a minor revision is required to be accepted.

Minor revision:

-          Chapter 2 begins with a general introduction and then moves on to paragraph 2.2. Paragraph 2.1 is missing. Please, check the subdivision.

      Thank you for bringing this to our attention.  We have corrected the subdivision of section 2.

-          Please, check the layout of the tables

We have adjusted the layout of the tables, while trying to conform to the journal requirements. The tables will likely need editorial adjustment to meet the format requirement of the journal.

Reviewer 3 Report

Comments and Suggestions for Authors

The authors of the present work described the role of STAT transcription factors in cancer and their use as drugs target.

The manuscript looks like well written and organized. The authors have presented an interesting topic in the field of oncology. The paper should be considered after major revisions.

1.       The figures have low resolution and are very difficult to understand. The authors should increase the words dimension and decrease the receptor illustrations;

2.       The authors should report the role of STAT transcription factors in the different kinds of cancer;

3.       Well-defined preclinical models are needed to better understand the role of STAT transcription factors in cancer diseases. Commercial cell lines cultured on common monolayer supports are in vitro systems not able to mimic the microenvironment of cancer diseases. 3D models and organ-on-chip represent some valuable research resources to reproduce the drug effect and sensitivity of tumors. For this reason, the authors should underline these aspects through a short overview of preclinical models in the field of cancer. The following references should be included in the manuscript: “Preclinical models in head and neck squamous cell carcinoma. doi: 10.1186/s41199-020-00056-4” and “Combining preclinical tools and models to unravel tumor complexity: Jump into the next dimension. doi: 10.3389/fimmu.2023.1171141”.

Author Response

The authors of the present work described the role of STAT transcription factors in cancer and their use as drugs target. 

The manuscript looks like well written and organized. The authors have presented an interesting topic in the field of oncology. The paper should be considered after major revisions.

  1. The figures have low resolution and are very difficult to understand. The authors should increase the words dimension and decrease the receptor illustrations;

We have modified the figures to reapportion the size of the graphics and increase the size of the text.

The authors should report the role of STAT transcription factors in the different kinds of cancer;

Well-defined preclinical models are needed to better understand the role of STAT transcription factors in cancer diseases. Commercial cell lines cultured on common monolayer supports are in vitro systems not able to mimic the microenvironment of cancer diseases. 3D models and organ-on-chip represent some valuable research resources to reproduce the drug effect and sensitivity of tumors. For this reason, the authors should underline these aspects through a short overview of preclinical models in the field of cancer. The following references should be included in the manuscript: “Preclinical models in head and neck squamous cell carcinoma. doi: 10.1186/s41199-020-00056-4” and “Combining preclinical tools and models to unravel tumor complexity: Jump into the next dimension. doi: 10.3389/fimmu.2023.1171141”.

We have included a new table (now Table 1) of well-defined preclinical models to better understand the role of STAT transcription factors in cancer diseases. We have also cited the two review papers that the reviewer listed.

Round 2

Reviewer 1 Report

Comments and Suggestions for Authors

This revised review article is about the expression and function of STAT transcription factors in physiological and cancer settings and their value as anticancer targets, well summarized by the group of leading experts of the STAT transcription factors. This reviewer believes that it has been improved a lot to be worth publishing and being read widely.

Reviewer 3 Report

Comments and Suggestions for Authors

Now, the manuscript is acceptable for a pubblication on "Cancers" journal